# Tech vs. Tradition: ChatGPT and Mindfulness in Enhancing Older Adults’ Emotional Health

**DOI:** 10.3390/bs14100923

**Published:** 2024-10-10

**Authors:** Ying Wang, Shiyu Li

**Affiliations:** 1Universal Design Institute, Zhejiang Sci-Tech University, Hangzhou 310018, China; lsy20000710@163.com; 2Zhejiang Provincial Key Lab for Healthy and Smart Kitchen System, China Academy of Art, Hangzhou 310020, China

**Keywords:** artificial intelligence, ChatGPT, mindfulness practice, older adults, negative emotions, life satisfaction

## Abstract

To improve older adults’ mental health, this study compared the effects of AI chatbots, such as ChatGPT 3.0, with traditional mindfulness therapies on loneliness and depression in older adults. Despite interest in AI as a complementary tool, empirical evidence on its impact remains limited. For an eight-week intervention, older adult participants from two nursing homes in Hangzhou were assigned to groups focused on mindfulness (group sessions) and chatting (one-on-one ChatGPT). Following the intervention, participants engaged in a researcher-led focus group discussion. After 8 weeks, tension had decreased significantly (*p* < 0.05) in the Mindfulness group of older adults, and there was no significant difference between the effects of ChatGPT and mindfulness on the emotional intervention of older adults. Findings indicated three themes, including (1) personal experience, reflecting older adults’ use of AI technology and mindfulness; (2) attitudes and perspectives on the experiment’s desirability and insufficiencies; and (3) needs and expectations for future AI and mindfulness developments, including usability and functional preferences. Similar to mindfulness practice, ChatGPT interactions helped older adults feel less depressed and might eventually reduce costs by replacing mindfulness. In the future, AI can be integrated with conventional techniques to improve interaction by giving AI a more human-like appearance.

## 1. Introduction

Global aging and technological advancements have prompted researchers to explore how technology can support aging populations and enhance older adults’ well-being. The increasing prevalence of empty nesters, driven by demographic shifts and adult children’s independent living, has become a focal point of public interest. Empty-nested older adult people are those who do not have children living with them, which includes both those who do not have children and those who live separately from their children [1]. In comparison to non-empty nesters, empty nesters experience a markedly greater sense of loneliness and exhibit a complicated range of psychological reactions [2]. According to an Indian study, people who are empty nesters tend to have higher depression symptoms [3]. According to a study on the mental health of older adults in China, over 34% of survey participants displayed depressive symptoms [4]. The causes of negative emotions among older adults are varied, including their own physical health [5], social support [6], economic status [7], and living environment [8].

Loneliness is a prevalent emotional distress among empty nesters [9,10], with one study reporting a mean loneliness score of 42.84 [11]. Loneliness occurs when an individual’s social expectations mismatch with their current situation [12]. The UCLA Loneliness Scale quantifies individual loneliness, with higher scores denoting greater loneliness. Loneliness in older adults is influenced by multiple factors, encompassing sociodemographic, physical, psychological, and cultural aspects, as well as social support [13]. Older adults who live in the outer suburbs are more likely to report loneliness than those who live in the city center. Older adults who live alone are more lonely than those who do not. Older adults with less education report being lonely more often than those with more. People with lower socioeconomic status reported more loneliness than older adults with higher socioeconomic status [14]. Physical pain and chronic illnesses exacerbate loneliness [15]. Insufficient social support correlates with loneliness [16], while social contact enhances mental health [17]. Loneliness can lead to negative consequences for older adults [18], including physical illness, depression, and suicide. It significantly lowers the quality of life and impairs social functioning in older persons who often lack coping mechanisms [19].

Studies indicate a high prevalence of depression among older adults [20], ranging from 28.5% to 28.8% [21]. These findings imply that depressive mood is a common issue among older adults. Depression is a negative emotional experience dominated by low mood [22] and is a common adverse emotion in the older adult population [23]. The presence of depressive symptoms in older adults can be measured by the Geriatric Depression Scale (GDS15). Many different and intricate elements can contribute to depression in older adults. Multiple factors contribute to depression in older adults, including lack of companionship in empty nesters [24], advanced age, marital status, poor finances, sleep quality, chronic diseases, and living conditions [25]. Sedentary lifestyles [26] and limited social participation [27] also correlate with increased depressive symptoms. Depressed mood can significantly impact older adults’ physical and mental health, potentially leading to life-threatening conditions [28]. It may disrupt sleep patterns [29], negatively affect physical health perceptions [30], and impair social functioning and quality of life [31].

Interventions for mood disorders in older adults include pharmacological treatments, psychotherapy, physical activity, social support, cognitive training, mindfulness practices, and chat-based care. For severe mood disorders like depression, medication is commonly prescribed to older adult patients. Various antidepressants are commonly used to treat geriatric depression [32]. In addition to medication, psychological intervention is also one of the most important means of treating geriatric depression. When used in conjunction with medication, psychological therapies can significantly improve the quality of life for older adults people suffering from depression [33]. Psychological relief therapy lowers sadness and anxiety by assisting patients in understanding and addressing the feelings and thought processes that contribute to depression [34]. In conclusion, psychological intervention and other non-drug therapies can be added to the medication regimen for older adult patients experiencing loneliness and depression in order to enhance both the therapeutic outcome and the patient’s quality of life. The medication regimen should be tailored to the individual needs of the patient and take into account their unique circumstances.

The improvement strategy for older persons experiencing depressive symptoms without a particular diagnosis should include social support, non-pharmacological therapy, emotional control techniques, and telemedicine care. Research has demonstrated that cognitive reappraisal techniques, which entail modifying one’s interpretation of occurrences and, as a result, the emotional reaction, can successfully lessen depressed symptoms [35]. Family support, hobbies, and physical activity are forms of social support associated with reducing depressive symptoms [25]. Individual nostalgia therapy can alleviate depression and reduce loneliness in geriatric internal medicine patients [36].

Mindfulness-based therapy is an intervention that has proven to be effective [37]. Mindfulness refers to a process that cultivates a state of mind characterized by non-judgmental awareness of the present moment, including one’s feelings, thoughts, bodily sensations, consciousness, and environment, while promoting openness, curiosity, and acceptance [38]. The fundamental premise of mindfulness practice is that experiencing the present moment without judgment and with openness can effectively counteract the effects of stressful stimuli [39]. Baer suggests that mindfulness therapy may be beneficial in treating anxiety and mood disorders [40]. Research indicates that mindfulness therapy is particularly effective in reducing anxiety and depressive symptoms in individuals with mood disorders, surpassing the placebo effect significantly [37]. In one systematic review, seven out of nine studies concluded that mindfulness therapy was advantageous for older adults experiencing anxiety or depression [41]. Another study found mindfulness therapy to be an acceptable approach for alleviating depressive symptoms in individuals aged 65 and older [42]. Therefore, this study will incorporate mindfulness as one of the interventions to enhance mood in older adults.

Chat nursing, a model well-suited to older adult patients’ characteristics, effectively alleviates depression among elderly inpatients [43] and plays an irreplaceable role in humanizing their care [44]. Traditional chat care, while providing timely feedback and emotional support, faces practical limitations such as labor costs and geographic constraints. Technology-based chatting offers a viable alternative. Smartphone apps and online platforms can overcome these limitations, enhancing flexibility and convenience in caregiver-older adult communication [45]. Artificial Intelligence (AI) develops computational systems that simulate human intelligence tasks. Generative AI, a subset relying on machine learning and deep learning, includes ChatGPT. This AI-powered chatbot provides tailored, intelligent conversational experiences by generating responses based on user input [46]. As a dialogue-based AI model, ChatGPT offers personalized, specialized, and real-time health information and support, surpassing traditional chat care. This is especially vital for older adults requiring close health monitoring [47]. ChatGPT offers unrestricted companionship for older adults, providing privacy and freedom of expression for shy individuals. However, it lacks genuine human interaction and emotional support. Technical barriers for some older adults may limit its accessibility. Consequently, ChatGPT’s effectiveness for older adults’ care requires further research. ChatGPT integrates current psychological intervention strategies, aiding nurses in scientific mental health assessments. Through personal accounts, it enables early identification and intervention for negative emotions via continuous interactions. By integrating multimodal data, ChatGPT provides nurses with comprehensive insights, enhancing the effectiveness of psychological interventions [48]. ChatGPT self-identifies as a potential source of companionship and comfort for isolated older adults through conversations and interactions. Research confirms that communication technologies can help alleviate loneliness and social isolation in older populations [49]. Advanced language models like ChatGPT show the potential to enhance social engagement and alleviate loneliness among older adults while reducing labor costs. Consequently, this study will employ ChatGPT companion chats as an intervention to improve mood in older adults.

Negative emotions such as loneliness and depression are common challenges faced by older adults, especially empty nesters, and can significantly impact their quality of life and social functioning. There are various programs, such as medication and psychotherapy, to deal with the problem of bad moods in older adults. The focus of this study is to compare the effectiveness of two modalities, ChatGPT Companion Chats and Mindfulness Exercises, in improving the negative moods of older adults, such as loneliness and depression. It is hoped that ChatGPT companion chats, as a novel technique, can alleviate negative moods in older adults. For older adults with medical conditions, ChatGPT companion chats may complement medication to enhance treatment effectiveness. Our research is motivated by a desire to contrast traditional, relatively specialized approaches with modern, technology-driven alternatives. Explore more possibilities for addressing mental health challenges in older adults. The study hypothesizes that both ChatGPT companion chats and mindfulness exercises could alleviate loneliness and depression in older adults, with no clear expectation of which intervention would yield superior results. Manual intervention usually requires trained professionals and is limited by time and place. This study hopes to explore the possibility of artificial intelligence replacing human intervention. Therefore, the focus of this study is to explore whether AI is effective in emotional intervention rather than to prove that ChatGPT is superior to mindfulness.

## 2. Materials and Methods

### 2.1. Study Design

The study used a controlled design to compare the effectiveness of ChatGPT Companion Chats and Mindfulness Exercises in improving loneliness and depression in older adults. Based on previous studies [50], an 8-week experiment was conducted in which participants were assigned to either the ChatGPT Group or the Mindfulness Group. Participants engaged in ChatGPT Companion Chats or Mindfulness Exercises for 15–30 min once a week. The researchers recorded participants’ mood, life satisfaction, and other relevant metrics at baseline (week 0) and at the end of the study (week 8).

### 2.2. Ethical Considerations

Ethical approval for the experiment was obtained from the Ethics Committee of Zhejiang Sci-Tech University. The study’s details were approved by the participating organizations’ management. Informed consent was acquired from participants or primary caregivers. Participants could withdraw at any time. All information was anonymized, participation was voluntary, and confidentiality was assured for both participants and collaborators.

### 2.3. Participants

Nursing homes provide a uniform environment for older adults, reducing sample heterogeneity and controlling for potential confounders in experimental settings. Researchers contacted two local nursing home directors to recruit participants. Inclusion criteria were age over 60, Mandarin communication ability, and normal cognitive function (MMSE score > 26) with basic mobility and communication skills.

### 2.4. Allocation to Interventions

Participants from two Hangzhou nursing homes were divided into ChatGPT and Mindfulness groups. Each group underwent weekly 30-min sessions for 8 weeks. Loneliness, depression, and life satisfaction were measured pre- and post-experiment. Post-experiment focus group discussions were conducted. Only participants completing all sessions were included in the data analysis.

#### 2.4.1. ChatGPT Group

Participants sat at a computer with a researcher providing assistance (Figure 1). The ChatGPT chatting method was explained, and both used headsets. Participants communicated verbally, with input converted to text and manually corrected if necessary. ChatGPT replies were presented as voice output to simulate live chat and reduce the reading burden. Researchers assisted with technology but did not interfere with chat content. The first session familiarized participants with the chat style, followed by seven 15–30 min one-on-one sessions.

#### 2.4.2. Mindfulness Group

A professional counseling expert-led participants in routine mindfulness practice, with researchers observing and assisting (Figure 2). Sessions followed the ‘Eight-Week Mindfulness Journey’ published by China Light Industry Publishing House, with varying content each week. Eight sessions were conducted in a one-to-many format. Specifically, this format refers to the structure where one facilitator (e.g., the mindfulness instructor) interacts with multiple participants simultaneously during the intervention sessions. The mindfulness sessions were conducted in a group setting, where the instructor guided the entire group of participants together.

#### 2.4.3. Focus Groups

Focus group interviews were conducted to gather supplementary insights. Two sessions were held following the conclusion of respective sub-experiments, each lasting approximately 30 min. The ChatGPT group comprised 2 researchers and 4 participants, focusing on the ChatGPT chaperone chat. The mindfulness group consisted of 2 researchers and 8 participants, centered on mindfulness practice. Researchers facilitated discussions as moderators and observers, respectively responsible for guiding topics and recording dialogue. The format included establishing rules, outlining objectives, in-depth discussions, and concluding remarks. Topics encompassed user experiences post-intervention, including prior usage, satisfaction, intervention impact, and interaction feedback. Researchers concluded sessions by expressing gratitude and encouragement to participants for their full participation, recognizing their adaptation to new technology, and ensuring a positive engagement experience.

### 2.5. Measurements

This study utilized five scales, namely the Geriatric Depression Scale 15-item version (GDS-15), the UCLA Loneliness Scale, the Brunel Mood Scale (BRUMS), the Satisfaction With Life Scale (SWLS), and the System Usability Scale (SUS).

#### 2.5.1. The Emotional State of Older Adults

Assessment of older adults’ emotional states utilized three mood scales. The Geriatric Depression Scale 15-item version (GDS-15), developed by Sheikh and Yesavage, gauged depression symptoms with 15 yes-or-no questions. Scores, ranging from 0 to 15, indicated severity: 0–4 (none), 5–9 (mild), 10–14 (moderate), and 15 (severe) depression. The UCLA Loneliness Scale by Russell et al., comprising 20 items, measured loneliness on a 4-point scale, with higher scores indicating greater loneliness. The Brunel Mood Scale (BRUMS) assessed older adults’ mood states via 32 items on a 4-point scale, delineating results across eight mood dimensions.

#### 2.5.2. Life Satisfaction

The Satisfaction With Life Scale (SWLS), developed by Pavot and Diener, was used to assess psychological satisfaction with life among older adults, which was a measure of subjective well-being. The scale comprises 5 items rated on a 7-point scale, with higher scores indicating greater satisfaction.

#### 2.5.3. System Usability

The System Usability Scale (SUS), developed by Brooke, was used to measure the perceived usability of a product or system. It consisted of 10 items and was divided into three subscales: ‘Effectiveness & Learnability’, ‘Use Efficiency & Usability’, and ‘Satisfaction’. The scale employed a 5-point rating system. The SUS usability score was calculated based on a formula, with higher scores indicating better system usability.

### 2.6. Data Analysis

The measurement data were subjected to statistical analysis using SPSS 27. Descriptive statistics and non-parametric tests were performed with SPSS. Since the normal distribution was not satisfied, the Mann–Whitney test was used for comparison between the two groups, and the Wilcoxon Rank-Sum Test was used for comparison between the two groups. Process data and focus group content were analyzed using thematic analysis with MAXQDA 2020 [51]. Audio data were transcribed, and initial coding of raw text data was performed. A coding library was developed, followed by grouping data in a second round of coding. Researchers identified categories and themes, refining final themes based on Renn’s adaptation [52].

## 3. Results

The chat companion experiment enrolled 15 participants, averaging 80.4 years old. Four females completed the process, averaging 88 years old, and were included in the final analysis. The mindfulness experiment involved nine participants, averaging 79.6 years old. Eight participants (seven females and one male) completed the process, averaging 78.6 years old, and were included in the final analysis.

### 3.1. Participant Characteristics

Table 1 shows the demographic variables and baseline results of all participants who completed the study. The majority of participants were female, with 11 (91.7%) female and 1 (8.3%) male. One participant (8.3%) had no formal education, six participants (50%) had primary school education, two participants (16.7%) had middle school education, two participants (16.7%) had high school education, and one participant (8.3%) had a university degree. Aside from the difference in educational background (*p* = 0.036 < 0.05), where the Chat group had a higher education level than the Mindfulness group, there were no significant differences between the two groups in other conditions. Ten participants reported mild depressive symptoms, and one participant reported moderate depressive symptoms.

Table 2 shows the changes in outcomes for the different groups. In the Mindfulness group, participants’ ‘tension’ emotions decreased after the experiment compared to before (1.88), and this difference was significant (*p* = 0.026 < 0.05). In the Chat group, participants’ ‘happiness’ emotions increased (3.25). Similarly, life satisfaction also increased (7.25). Life satisfaction in the Mindfulness group also increased (4.5).

From the results, both groups of participants showed a certain degree of decrease in negative emotions (anger, depression, fatigue, confusion). There was also a certain degree of increase in vitality, happiness, and life satisfaction, but loneliness also increased. The participants in the mindfulness group showed a decrease in depression scores. Among them, one participant transitioned from moderate to mild depressive symptoms, and one participant transitioned from mild to moderate. There were no significant between-group differences in any of the data, suggesting to some extent that there is no difference in effectiveness between ChatGPT chaperone chat and mindfulness practice.

### 3.2. Thematic Analysis

Two focus group sessions were conducted, with four participants engaging in discussions on ChatGPT chaperone chat and eight participating in discussions on mindfulness practice. Qualitative analysis of the chat content and focus group discussions revealed three main themes and their subcategories (Table 3), with representative quotes explaining each category.

#### 3.2.1. Personal Experience

All participants reported no exposure to ChatGPT. However, some participants mentioned similar experiences with AI contact. For instance, subject d (84 years old) remarked, ‘Sometimes when I go to the hospital with a cold, there will be a robot, and you can ask it which ward is on what floor. Which floor should I go to if I want to see a doctor’. The conversational pattern of the hospital guide robot bears some resemblance to the experimental intelligent companion chat, which brought a sense of familiarity to older adults. The study suggests that older adults find the use of AI technology acceptable. They perceived that service robots could assist them in performing their daily tasks more effectively [53]. Some participants had prior exposure to mindfulness or similar exercises, such as meditation. Subject k (78 years old) commented, ‘I’ve learned some relaxation and meditation techniques at the community center before, but I haven’t practiced them systematically’.

Several participants indicated that they have experience with smartphones, and some exhibit a strong dependency and fondness for them. Subject c (92 years old) mentioned in the chat, ‘My daughter took my smartphone away, which makes me quite angry right now. I am really sad inside without it. Since I’ve had a smartphone for 14 years, I will not use a senior phone. When I’m without my smartphone, I get sad’. Studies reveal a favorable relationship between the amount of time spent on smartphones and smartphone addiction [54]. Subject c (92 years old) was the participant who had been using a smartphone for a long time and showed a strong need for it. Another participant mentioned, ‘I use my phone to read the news and watch wellness programs, especially medical-related ones. It’s important to have some understanding, and I enjoy it’. This suggests that older folks, who can also use smart technology and benefit from its convenience, have found greater ease in their lives as a result of its advancements. However, some older adults said that they feel unfamiliar with artificial intelligence because they do not have a smartphone but only an old-age phone. In a similar vein, some older adults have never heard of mindfulness. Subject h (85 years old) stated, ‘I have never come into contact with mindfulness; this is kind of my first experience and I find it quite refreshing’. 

#### 3.2.2. Attitudes and Perspectives

Some older adults who experience a chaperoned conversation say they benefit from it and will prepare for it or look forward to it before they have it. Subject d (84 years old) would set a theme or question for herself before each conversation, and the conversation would revolve around the theme. She said during the discussion: ‘Every conversation, I thought of a question and it would give me a more comprehensive answer than what I had thought of on my own. It knows a lot about what’s going on in life. Once I asked it about its views on genetically modified foods and its answer was not one-sided. I think that’s still instructive’. Many older adults expressed positive attitudes towards mindfulness. A subject said, ‘I feel that mindfulness practice has not only brought me peace and relaxation mentally but has also had a positive impact in other aspects of my life. For example, I find myself paying more attention to healthy eating and getting enough sleep, as well as paying more attention to the good things around me’. And the subject said, ‘I think mindfulness practice has made me appreciate each day more’. Participants in both groups saw their life satisfaction scores rise after the intervention, and the upward trend was close to significant.

There were also older adults who gradually began to change their attitudes as the chatting progressed. Subject a (88 years old) conversations started with ‘no’, ‘I have nothing to say’, and ‘there is nothing I want to ask’. After that, she started to chat and would take the initiative to greet, ‘Old friend, we meet again, how are you’. The change in response also reflects the change in mood and emotion of older adults. Older adults grew more laid back and gregarious as the number of interactions rose. Some older adults also said: ‘I feel better after chatting’. Additionally, compared to non-users, internet users reported feeling more support from friends and family [55]. On the other hand, participants in the mindfulness group mentioned certain challenges they encountered. ‘During the experiment, sometimes I would feel that I lacked motivation and patience and it was hard to keep going’. ‘Sometimes my thoughts would wander to other places and it was hard to concentrate’. 

Some of the older adults involved in the chat group expressed that the experimental format was less engaging and that the requirement for standard Mandarin in dialog recognition posed challenges. For example, subject d (84 years old) said: ‘Young people speak good Mandarin, but my generation didn’t have pinyin or anything like that when we were studying, so we usually spoke dialects. I’m from Ningbo and speak the Ningbo dialect, so it doesn’t understand this dialect’. Another subject said, ‘Sometimes the words I say are not understood by it, and you have to adjust them on the side, or the words that appear differ from what I say’. The care home’s director remarked, ‘This companion talk feels a touch too robotic. It’s quick and rigid. Older adults are bored and unable to use computers when they are using them’. Research indicates that older adults experience declines in language comprehension with age, leading to poorer performance compared to younger counterparts in understanding complex conversations [56]. Hence, there is a need for further refinement of the current form of accompanied conversation to enhance older adults’ comprehension and utilization of this technology.

#### 3.2.3. Needs and Expectations

Numerous individuals have shared their opinions and views on AI technology. Older adults typically communicate online through voice or handwritten text, so visual and auditory aspects are highly valued by them during accompanied conversations. Some older adults may prefer forms that imitate real beings, such as charming and cartoonish designs, like little animals. Robots are not very popular. A subject said: ‘If the robot is a very cute shape, like a Barbie doll or a big monkey, it will also offer to say hello. That kind of is very likable’. Some participants indicated that sight and hearing include not only appearance but also mannerisms and tone of voice, and some expressed a preference for interacting with real people. The director of the nursing home said: ‘Older adults prefer their children to come and see them offline when they live here, not just through a telephone greeting. They prefer real contact. Older adults also rarely use video calls’. Additionally, prior research has demonstrated that older persons still prioritize face-to-face communication above other forms of communication, even despite the rapid advancement of online communication tools [57].

Researchers observed the impact of hearing loss on older adults during both the process and interviews. Due to age-related decline, their ability to use voice-based systems was affected [58], necessitating devices with higher volume. During the experiment, some older adults opted to read text responses because they could not hear the device’s voice replies clearly. Some subjects said, ‘It’s not that they don’t want to use a smartphone, but senior phones are simpler and loud’. Thus, auditory design is a point of concern for products used by older adults.

Participants in the mindfulness group showed a highly positive attitude towards this format, expressing a desire for deeper understanding and the ability to apply it in their daily lives. Subject g (78 years old) said, ‘I learned some practical skills and strategies through this experiment. I hope I can better apply mindfulness practice in my future life’. Subject l (80 years old) said: ‘I think mindfulness has helped me with my emotional management. I hope I can still learn more to apply it to my life and improve my quality of life’. Furthermore, studies have demonstrated that mindfulness training can improve the mental and physical health of older adults by teaching them to accept and cope with chronic illnesses more effectively [59].

## 4. Discussion

Overall, we discovered that individuals’ feelings of depression and loneliness did not significantly change as a result of either ChatGPT companionship or mindfulness practice. Nonetheless, following the intervention, the data showed a decrease in negative emotions in both groups, with no discernible variations between the groups. This inadvertently implies that ChatGPT can function similarly to mindfulness in mitigating negative emotions in older adults and that both companionship and mindfulness have similar benefits in decreasing negative emotions in older adults.

In particular, after the trial, Subject h (85 years old) depression score dropped from 11 (moderate depressed symptoms) to 5 (mild depressive symptoms). He said he ‘felt much better mentally, and the worries were not as heavy’ following the practice. This is similar to Subject g (80 years old), ‘performing these exercises helped my mind feel extremely relaxed; I used to overthink a lot’. Subject d (84 years old) saw a decrease in her depression level in the ChatGPT group as well. ‘Chatting with ChatGPT was quite enjoyable; it felt like catching up with an old friend’, the woman said.

After the trial, the loneliness score of the 85-year-old individual increased from 4 to 13. ‘I used to think about the past when I meditated, and now I feel more and more alone’, she said in her explanation. In contrast, there were notable individual variances even while the ChatGPT group’s aggregate level of loneliness rose. For instance, the loneliness score of subject b (88 years old) dropped from 24 to 7. ‘ChatGPT feels like my little companion, someone I can talk to’, she said with a smile.

There could be several factors contributing to these outcomes. First of all, medical issues, social support, and familial ties are just a few of the variables that can impact depression and lonely experiences in older persons [60]. It may not be possible to achieve notable benefits with a single, brief session. Second, research from the past suggests that emotional state modifications take time [61], so the 8-week intervention period may not be long enough to see the full benefits. Individual psychological adjustment capacities may also have an effect on the results. For instance, Auntie Cheng, who finished the study, saw a decline in her physical well-being and a reduction in family visits, both of which added to her depressive feelings.

During the experiment, a few notable situations were noted. There was no visual interface during the technology-assisted chat sessions, so communication was restricted to text and speech. Both visual attractiveness and engagement suffered as a result. A higher dropout rate was also caused by the older adults’ decreased capacity to adopt new technology. However, several participants thought that mindfulness exercises were abstract. ‘I don’t entirely comprehend the importance of those behaviors’, said subject j (74 years old). It is interesting that in the chat group, the four who completed the process are nearly 8 years older than the average for the 15 that started. Older participants may have had more time or motivation to complete the intervention, which could be a factor worth exploring in future studies. One of the limitations of the study is the relatively small and unbalanced sample sizes, with 15 participants in the ChatGPT group and 9 in the mindfulness group. This may limit the generalizability of the results. We observed several compliance-related challenges during the study. Some participants struggled to maintain consistent engagement with the 8-week intervention. For example, some participants stopped participating midway because they went to the hospital for treatment, and some participants refused to continue because they were not willing to express themselves. Future research could benefit from larger, more balanced samples to improve statistical power and the reliability of the findings.

In conclusion, despite the little overall improvement, the majority of participants expressed a greater sense of life satisfaction and gave the experimental forms positive feedback. This result is consistent with other research that indicates mindfulness techniques [62] and virtual communication [63] can provide older persons with social support and coping mechanisms, which can improve their quality of life [64]. Thus, we think that in the future, offering older adults a variety of easy social and psychological support channels through the use of technology and psychological interventions would be crucial. The study also shows that older persons have distinct preferences for the types and contents of interventions, highlighting the need for more individualized and humanized service solutions that cater to the needs of various populations.

## 5. Conclusions

The purpose of this study was to look at how mindfulness training and ChatGPT companionship affected depressive and lonely feelings in older persons. We discovered that, in the short term, both ChatGPT companionship and mindfulness meditation had rather limited impacts on significantly reducing depression symptoms and feelings of loneliness in older persons based on trials conducted with elderly patients in nursing homes.

It is noteworthy, nevertheless, that both groups of older persons exhibited some degree of reduction in negative emotion scores, even if there was no fundamental shift in emotions. This implies that, in contrast to conventional mindfulness therapies, cutting-edge artificial intelligence-based interactive modes like ChatGPT might also be useful for reducing negative emotions and enhancing the mental health of older adults.

Actually, giving older adults a technique to relax their bodies and minds and concentrate on the here and now helps them change their attention and control their emotions, which is largely why mindfulness interventions are so successful. Even though ChatGPT takes a different tack, it nonetheless gives older adults a chance to engage with the outside world by mimicking human speech. This gives them a brief break from an inactive, alone existence, which in turn promotes some level of relaxation and comfort.

Thus, the results of this study offer important new information about how the new technology ChatGPT can improve older adults’ feelings. Investigating how to more effectively use intelligent tools to meet the social and psychological needs of older adults and achieve a positive interaction between humans and machines will be an important topic worth looking into as artificial intelligence technology continues to develop and improve. To improve affinity and attraction, AI partners may, in the future, be equipped with more personalized and human designs. In addition, older adults can have complete companionship experiences in immersive virtual environments made possible by VR/AR technologies.

## Figures and Tables

**Figure 1 behavsci-14-00923-f001:**
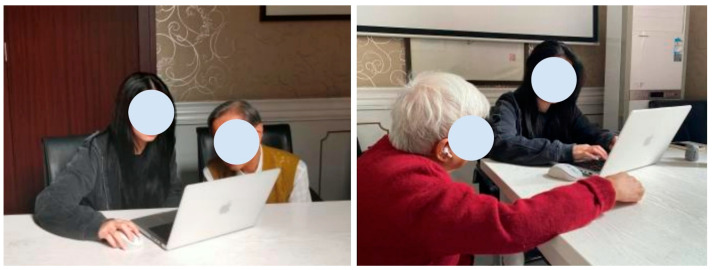
ChatGPT group.

**Figure 2 behavsci-14-00923-f002:**
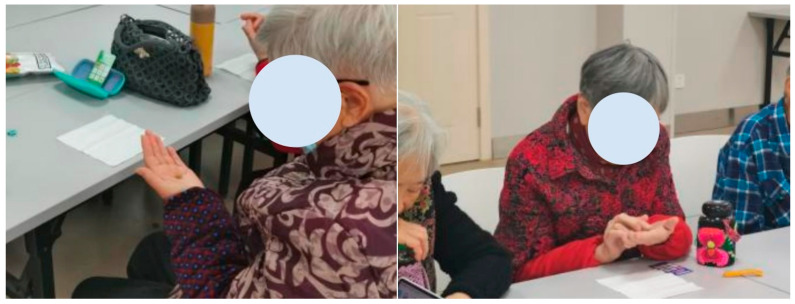
Mindfulness group.

**Table 1 behavsci-14-00923-t001:** Participant characteristics (n = 12).

Variable		Chat (n = 4)			Mindfulness (n = 8)			*p*
		n (%)	Median	Q1	n (%)	Median	Q1	
Gender	Male	4 (100.0)			7 (87.5)			0.480
	Female	0 (0)			1 (12.5)			
Educational background	not attending school	0 (0)	5.0	2.8	1 (12.5)	2.0	2.0	0.036 *
	secondary schools	1 (25.0)			5 (62.5)			
	Junior high school	0 (0)			2 (25)			
	three-year college	0 (0)			0 (0)			
	High school	2 (50.0)			0 (0)			
	University	1 (25.0)			0 (0)			
	postgraduate education	0 (0)			0 (0)			
GDS-15 score			6.0	3.8		7.0	6.0	0.146
UCLA score			12.0	7.8		14.0	4.8	0.932
BRUMS score	anger		2.5	1.3		2.5	0.3	1.000
	tension		3.0	1.3		1.5	0.3	0.227
	depression		1.5	0.3		1.0	0.0	0.663
	vigor		4.5	0.3		9.0	7.3	0.119
	fatigue		7.5	2.8		3.0	1.3	0.147
	confusion		4.0	2.3		1.0	0.3	0.086
	happy		7.00	4.8		11.0	10.0	0.057
	calmness		11.5	7.0		12.0	9.3	0.665
SELE score			22.5	19.3		27.5	26.0	0.230

* *p* < 0.05.

**Table 2 behavsci-14-00923-t002:** Before and after differences between the two groups.

	Chat (n = 4)					Mindfulness (n = 8)					Change in *p*
	Median1	Median2	*p*1	Change in Mean	Change in Median	Median1	Median2	*p*2	Change in Mean	Change in Median	
GDS-15 score	6.0	6.0	1.000	0	0.0	7.0	7.0	1.000	−0.38	0.0	0.930
UCLA score	12.0	10.0	0.715	1.00	1.5	14.0	15.0	0.888	1.13	2.5	0.932
BRUMS score											
anger	2.5	1.0	1.000	−0.25	−1.0	2.5	0.0	0.061	−1.88	−2.0	0.259
tension	3.0	0.5	0.197	−0.25	−2.5	1.5	0.0	0.026 *	−1.63	−1.0	0.492
depression	1.5	1.0	0.705	−1.25	0.0	1.0	0.0	0.176	−1.00	−0.5	0.796
vigour	4.5	6.5	0.197	2.00	2.0	9.0	9.5	0.865	0.38	0.5	0.260
fatigue	7.5	2.5	0.144	−5.00	−6.0	3.0	0.0	0.106	−2.50	−2.5	0.307
confusion	4.0	1.0	0.197	−3.50	−3.5	1.0	0.0	0.443	−1.13	−0.5	0.391
happy	7.0	10.0	0.066	3.25	2.5	11.0	13.0	0.306	1.50	2.0	0.493
calmness	11.5	10.0	0.705	−1.25	0.0	12.0	11.5	0.725	0.50	0.0	0.492
SELE score	22.5	30.5	0.068	7.25	8.0	27.5	32.0	0.051	4.50	3.5	0.269

* *p* < 0.05, mean1 represents before intervention and mean2 represents after intervention.

**Table 3 behavsci-14-00923-t003:** Themes and categories reported by participants after the focus group.

Themes	Categories
personal experience	Experience with AI or mindfulness
	There is a dependence on smartphones
	New to AI/mindfulness
Attitudes and perspectives	There are benefits to chatting with ChatGPT or mindfulness practice
	Change in mood
	Do not think this is a good form of AI
Needs and expectations	Prefer bionic-looking forms, real human contact
	Impact of hearing deterioration
	I hope to be able to use it in my daily life

## Data Availability

The data presented in this study are available on request from the corresponding author due to privacy.

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
