# Peer review of "Tech vs. Tradition: ChatGPT and Mindfulness in Enhancing Older Adults’ Emotional Health"

_behavsci, 2024, doi:10.3390/bs14100923_

Round 1
Reviewer 1 Report
Comments and Suggestions for Authors
Dear Authors,
Thank you for your submission. It seems the paper needs some major/critical updates in order to be sound.
In the abstract it is stated "...After 8 weeks, tension had decreased significantly (p<0.05) in the Mindfulness group of older adults, and there was no significant difference between the effects of ChatGPT and mindfulness on the emotional intervention of older adults. ..." However, in the ending paragraph of the introduction it says "...The main research hypotheses were that both ChatGPT companion chats and mindfulness exercises could alleviate loneliness and depression in older adults, with ChatGPT companion chats expected to yield superior results. ..." According to your results, this does not seem to be the case. The statement in the introduction is a contradicting one.
The sample sizes seem a bit small and might be unbalanced. For example there are 15 participants in the chat group but 9 in the mindfulness. Would it not be more meaningful to score the effectiveness of ChatGPT and mindfulness approaches, if both of these test groups involved the assessment of the same individuals?
It is written "... The measurement data were subjected to statistical analysis using SPSS. ..." But which statistical model is used? It is not defined in the paper and is important. The models need to be clearly defined and described.
Your data is non-parametric and therefore needs a non-parametric test (e.g. Wilcoxon or etc.). Since the variables appear to be in ordinal scale the central measure statistics should be given as medians. However, it appears to be treated parametric and i assume you used t-testing. So, it looks like there is a mismatch of data and tests.
I am also a bit skeptical of the comparison of ChatGPT and mindfulness as you know, they are very different things. If i did not forget, this was already mentioned in the paper but it could be useful to elaborate a bit more and justify why they are compared. They could both help with depressive symptoms but are there any other reasons to choose the two for comparison? For example, it would make a big justification if it is shown that the mindfulness is a common approach used to help the elderly.
Please correct the citation [60] as it is missing author names.
Thanks
Comments on the Quality of English Language
Just a proof reading, a regular spell/grammar check should do ok.
Author Response
Thank you very much for taking the time to review this manuscript. Please find the detailed responses below and the corresponding revisions/corrections highlighted/in track changes in the re-submitted files.
Comments 1: In the abstract it is stated "...After 8 weeks, tension had decreased significantly (p<0.05) in the Mindfulness group of older adults, and there was no significant difference between the effects of ChatGPT and mindfulness on the emotional intervention of older adults. ..." However, in the ending paragraph of the introduction it says "...The main research hypotheses were that both ChatGPT companion chats and mindfulness exercises could alleviate loneliness and depression in older adults, with ChatGPT companion chats expected to yield superior results. ..." According to your results, this does not seem to be the case. The statement in the introduction is a contradicting one.
Response 1: Thank you for pointing out this inconsistency. We apologize for our misstatement. We have rewritten this element of the text.
Page 5, line 155: The study hypothesizes that both ChatGPT companion chats and mindfulness exercises could alleviate loneliness and depression in older adults, with no clear expectation of which intervention would yield superior results.Manual intervention usually requires trained professionals and is limited by time and place. This study hopes to explore the possibility of artificial intelligence replacing human intervention.
Comments 2: The sample sizes seem a bit small and might be unbalanced. For example there are 15 participants in the chat group but 9 in the mindfulness. Would it not be more meaningful to score the effectiveness of ChatGPT and mindfulness approaches, if both of these test groups involved the assessment of the same individuals?
Response 2: Thank you for your insightful comment regarding the sample sizes and the suggestion to assess both interventions using the same individuals.
We acknowledge that the sample sizes of the ChatGPT group and mindfulness group were somewhat small and uneven due to participant compliance issues, which may limit the statistical power and generalisability of the findings. Specifically, some participants had difficulty maintaining regular attendance during the 8-week intervention, primarily due to scheduling conflicts and accessibility challenges. For example, some older adults interrupted the participation process due to health reasons, and some older adults arranged to meet with relatives and cancelled the participation. In the revised manuscript, we address this limitation in the discussion section. And suggests that future studies could benefit from a larger, more balanced sample.
Page 16, line 450: One of the limitations of the study is the relatively small and unbalanced sample sizes, with 15 participants in the ChatGPT group and 9 in the mindfulness group. This may limit the generalizability of the results. We observed several compliance-related challenges during the study. Some participants struggled to maintain consistent engagement with the 8-week intervention. For example, some participants stopped participating midway because they went to the hospital for treatment, and some participants refused to continue because they were not willing to express themselves. Future research could benefit from larger, more balanced samples to improve statistical power and the reliability of the findings.
As for the suggestion of using a within-subjects design where the same individuals undergo both interventions, this is an excellent point. However, for this study, we chose a between-subjects design to avoid potential carry-over effects that might arise if participants were exposed to both interventions in succession. Emotion is an irreversible variable.This approach was intended to ensure that each intervention could be evaluated independently, without interference from prior experiences with the other intervention.
Thank you again for your valuable feedback, which has helped us improve the clarity and rigor of our manuscript.
Comments 3: It is written "... The measurement data were subjected to statistical analysis using SPSS. ..." But which statistical model is used? It is not defined in the paper and is important. The models need to be clearly defined and described.
Response 3: Thank you for pointing out our oversight in the manuscript. Based on your feedback, we have revised the methods section to provide a more detailed description of the statistical model applied.
Page 8, line 254: Descriptive statistics and non-parametric tests were performed with spss. Since normal distribution was not satisfied, Mann-Whitney test was used for comparison between the two groups, and Wilcoxon Rank-Sum Test was used for comparison between the two groups.
Comments 4: Your data is non-parametric and therefore needs a non-parametric test (e.g. Wilcoxon or etc.). Since the variables appear to be in ordinal scale the central measure statistics should be given as medians. However, it appears to be treated parametric and i assume you used t-testing. So, it looks like there is a mismatch of data and tests.
Response 4: Thank you for your valuable comment. We appreciate your observation regarding the nature of our data and the need for appropriate statistical methods.Upon review, we acknowledge that the data are indeed non-parametric.We have revised the statistical analysis in the manuscript to reflect the use of non-parametric tests. In addition, we have updated the central tendency measurement reported as the median in the Table1 and Table2.Thank you again for your insightful feedback. We believe this revision significantly enhances the methodological rigor of the manuscript.
Page 8, line 254: Descriptive statistics and non-parametric tests were performed with spss. Since normal distribution was not satisfied, Mann-Whitney test was used for comparison between the two groups, and Wilcoxon Rank-Sum Test was used for comparison between the two groups.
Comments 5: I am also a bit skeptical of the comparison of ChatGPT and mindfulness as you know, they are very different things. If i did not forget, this was already mentioned in the paper but it could be useful to elaborate a bit more and justify why they are compared. They could both help with depressive symptoms but are there any other reasons to choose the two for comparison? For example, it would make a big justification if it is shown that the mindfulness is a common approach used to help the elderly.
Response 5: Thank you for your thoughtful comment and for raising this important concern.We agree that ChatGPT and mindfulness are different interventions. However, they were chosen for comparison in this study because both approaches have have shown promise in potentially reducing depressive symptoms, particularly in older adults.
Mindfulness exercises are widely recognized as a non-pharmacological method for improving mental well-being in the elderly, with a strong evidence base supporting its efficacy in reducing stress, anxiety, and depression. In fact, mindfulness is commonly employed in clinical and therapeutic settings for older adults as part of holistic mental health interventions.We use previous literature in the manuscript to support this.
Page 3, line 107: In one systematic review, seven out of nine studies concluded that mindfulness therapy was advantageous for older adults experiencing anxiety or depression [41]. Another study found mindfulness therapy to be an acceptable approach for alleviating depressive symptoms in individuals aged 65 and older [42].
On the other hand, ChatGPT companion chats represent an emerging, technology-based intervention that leverages conversational AI to provide social interaction and emotional support. Given the increasing role of technology in supporting mental health, it was of interest to explore whether such an innovative tool could offer comparable emotional and psychological benefits to a well-established method like mindfulness.Mindfulness is more professional.If ChatGPT can achieve a similar effect to mindfulness, it has found a possibility that is easier to achieve and less costly to human labor.
We have supplemented the manuscript to further illustrate the reasons for comparing the two measures.
Page 4, line 152: Our research is motivated by a desire to contrast traditional, relatively specialized approaches with modern, technology-driven alternatives. Explore more possibilities for addressing mental health challenges in older adults.
Comments 6: Please correct the citation [60] as it is missing author names.
Response 6: Thank you for bringing this to our attention. This has been corrected in the revised manuscript.
Reviewer 2 Report
Comments and Suggestions for Authors
Please do not use the term “elderly” in your paper. In this document you use the term elderly, the term senior, senior citizen, the term older people, the term older individuals, and the term older adults. There may be other terms you use as well. Please use only the term “older adults.” Remove all other terms to refer to the population. The American Psychological Association recommends the use of the term “older adults.” Please utilize only that term throughout your paper when referring to the older adult population. The term “elderly” is ageist.
Introduction
I am wondering, what role the fact that the mindfulness intervention involved a group as opposed to one on one might have had.
“Rural, childless, widowed, older, less educated, and lower-class elderly have higher loneliness rates [14].” P.2 line 45- 46 I think you mean lower socioeconomic status. Be sure to state it that way. Also, you do not provide a comparison group. For instance, do you mean rural, as compared to urban or suburban or both? Widowed as compared to married? Widowed as compared to divorced? You need to provide some clarification and comparisons in this sentence. Look back at the sources to see what they were comparing these statuses to.
“Various antidepressants, including escitalopram, sertraline, venlafaxine, mirtazapine, paroxetine, and others, are commonly used to treat geriatric depression [32].” P.2 line 68 - 69. Please remove the list of drugs as they vary in name from country to country and the actual names are not important for what you are discussing here.
You did a solid job in your introduction in validating the need for your research.
Materials and methods
Study design
I am concerned that 8 15 to 30 minute interventions once a week for eight weeks is not sufficient of an intervention to make any kind of claims. It is just too small of an intervention.
On the bottom of page 4 line 185 for you describe a “one-to-many format“ please provide additional details as to what that means
I like your use of focus groups as a post evaluation strategy.
Measurements
You need to use consistent verb tensing when discussing the various instruments utilized. For instance, you stated,
“2.5. Measurements
This study will utilize five scales, namely the Geriatric Depression Scale…” then later you stated, “Assessment of older adults' emotional states utilized three mood scales.” I am not sure what the verb tensing requirements are in the citation format you are using but minimally you should keep a consistent verb tensing for each of the various sections of the measurement tools utilized.
Results
In addition to a very limited intervention, you also had very low numbers. If I am reading this correctly, your chat group had 15 participants, but only four completed the process. The mindfulness experiment had nine and eight completed the process. Those seem like very low numbers for making any kind of implications. It is interesting that in the chat group the four that completed the process are nearly 8 years older than the average for the 15 that started.
“There were no significant between-group differences in any of the data, suggesting to some extent that there is no difference in effectiveness between ChatGPT chaperone chat and mindfulness practice.” P.7 lines 261-263. I can’t help but wonder if the fact that these were chaperoned chats had an impact here. That seems more of a social interaction than traditional AI chats. I realize that the population served in the study likely could not chat without assistance, but still, how much of this is the social experience of being with a chaperone.
I have a lot of concerns about your quantitative results since there was such a limited intervention for both the mindfulness and the chat AI experiences. My suggestion would be to increase the intervention significantly over a longer period of time, increase the number of participants, and look for ways to determine the actual causes of the changes rather than assuming they are simply from either the mindfulness or the chat AI experiences.
“Participants in both groups saw their life satisfaction scores rise after the intervention, and the upward trend was close to significant.” P.9 315–3 17. You cannot state that something is “close to significant” if it is not significant, then any changes could be based on things other than the intervention. Please remove that part of the sentence.
“The disadvantages of the Chat group also exist, the experimental format is boring for
the elderly. The dialog recognition requires more standard Mandarin, which is very difficult for the elderly.” P.9 330–332. This sounds like a pretty bold generalization. Perhaps you could state it as “some of the older adults involved in the chat group expressed…”
“Elderly people like forms that imitate real beings, such as charming and cartoonish designs, like little animals.” page 9 lines 348–349. Here is another example of a bold generalization. In the line before that you say “elderly individuals typically communicate…“ At least that leaves the possibility that others may use other types of communication. Be careful about bold generalizations in your comments here. You cannot say all older adults do anything. You do not have the data to support that.
“Sight and hearing not only include appearance, but also mannerisms and tone of voice, and the elderly prefer to be in contact with real people” Page nine lines 351 to 352. Here is another example of generalization. Please go back through your results section and rephrase so you are not trying to indicate that all older adults have these experiences but only a subset of some of the people that you spoke with in this research.
Your focus group results do not seem to be addressing the question at hand. I think there are many interesting pieces of information that you drew from the focus group, but as far as the question of the effectiveness of ChatGPT and mindfulness on minimizing depressive and loneliness symptoms, I really don’t feel like you got there with what you provided here.
The paragraph on page 10 from nine 395 to line 403 really provides an excellent overview of the limitations of your study. My suggestion would be to increase the intervention length both in the length of time for the individual interventions and the number of times the interventions occur. In other words, I think if you did this twice a week over a four month period of time and increased your participant size to include many more people you might have significant findings that would be much more valuable than what you have here.
My recommendation is to use the work here to design a more impactful intervention and evaluate it. I do not recommend that you publish this work. There are just too many limitations.
no feedback
Author Response
Thank you very much for taking the time to review this manuscript. Please find the detailed responses below and the corresponding revisions/corrections highlighted/in track changes in the re-submitted files.
Comments 1: Please do not use the term “elderly” in your paper. In this document you use the term elderly, the term senior, senior citizen, the term older people, the term older individuals, and the term older adults. There may be other terms you use as well. Please use only the term “older adults.” Remove all other terms to refer to the population. The American Psychological Association recommends the use of the term “older adults.” Please utilize only that term throughout your paper when referring to the older adult population. The term “elderly” is ageist.
Response 1: Thank you for your insightful comment regarding the terminology used in the manuscript.We agree with your suggestion and acknowledge the importance of using respectful and inclusive language. As recommended, we have revised the entire manuscript to consistently use the term "older adults" when referring to the population. All instances of "elderly," "senior," "senior citizen," "older people," and "older individuals" have been replaced with "older adults" to align with the guidelines set by the American Psychological Association and to ensure the manuscript uses non-ageist language.
Comments 2: Introduction. I am wondering, what role the fact that the mindfulness intervention involved a group as opposed to one on one might have had.
Response 2: Thank you for raising this important point. The group format of the mindfulness intervention could indeed have played a significant role in the outcomes.
We refer to previous relevant studies, and different studies have shown that group and individual therapy is effective in improving the adverse mood of older adults.(DOI:10.3969/j.issn.1672-9455.2021.11.023;DOI:10.16440/j.cnki.1674-8166.2020.06.011;DOI:10.19792/j.cnki.1006-6411.2022.26.028) Because group mindfulness training can affect multiple individuals at the same time, it is suitable for situations that need to be widely promoted and quickly implemented, especially in improving mental health and interpersonal relationships. One-on-one mindfulness training, on the other hand, is more appropriate for situations that require personalized attention or deep involvement.For this study, group training is more suitable.
Mindfulness in a group may provide additional social interaction and support, which can positively influence emotional well-being. This sense of belonging and shared experience may enhance the effectiveness of the intervention by addressing feelings of loneliness and promoting peer connection.
In the revised manuscript, we have expanded the description of the "one-to-many format" used in our study to help readers better understand it..
Page 6 ,line 207: Specifically, this format refers to the structure where one facilitator (e.g., the mindfulness instructor) interacts with multiple participants simultaneously during the intervention sessions. The mindfulness sessions were conducted in a group setting, where the instructor guided the entire group of participants together.
Comments 3: “Rural, childless, widowed, older, less educated, and lower-class elderly have higher loneliness rates [14].” P.2 line 45- 46 I think you mean lower socioeconomic status. Be sure to state it that way. Also, you do not provide a comparison group. For instance, do you mean rural, as compared to urban or suburban or both? Widowed as compared to married? Widowed as compared to divorced? You need to provide some clarification and comparisons in this sentence. Look back at the sources to see what they were comparing these statuses to.
Response 3: Thank you for your detailed feedback. Upon revisiting the sources, we have clarified the specific comparisons made in the original research. We have revised the sentence accordingly to make these comparisons explicit.
Page 2 ,line 48: Older adults who live in the outer suburbs are more likely to report loneliness than those who live in the city centre. Older adults who live alone are more lonely than those who don't. Older adults with less education report being lonely more often than those with more.People with lower socioeconomic status reported more loneliness than older adults with higher socioeconomic status [14].
Comments 4: “Various antidepressants, including escitalopram, sertraline, venlafaxine, mirtazapine, paroxetine, and others, are commonly used to treat geriatric depression [32].” P.2 line 68 - 69. Please remove the list of drugs as they vary in name from country to country and the actual names are not important for what you are discussing here.
Response 4: Thank you for your valuable feedback. We agree that the specific names of the antidepressants are not essential to the overall discussion, and we acknowledge that drug names can vary internationally.In response to your suggestion, we have removed the list of specific antidepressants from this section.
Page 3 ,line 75: Various antidepressants are commonly used to treat geriatric depression [32].
Comments 5: Materials and methods. Study design. I am concerned that 8 15 to 30 minute interventions once a week for eight weeks is not sufficient of an intervention to make any kind of claims. It is just too small of an intervention.
Response 5: Thank you for raising this concern regarding the duration and frequency of the interventions.We understand that the intervention period might seem limited for making substantial claims about effectiveness.To address this concern, we have considered participant adherence and the practicalities of implementing the intervention. Previous studies have shown that 8-week simple mindfulness training can effectively reduce negative emotions such as anxiety and depression in older adults.(DOI: 10.11569/wcjd.v28.i7.265)The choice of 15 to 30-minute sessions, once a week for eight weeks, was based on balancing feasibility with participant engagement and adherence. Longer or more frequent sessions could have potentially impacted participant retention and adherence.The fact shows that participant compliance is indeed a difficult problem to solve.
Moreover, we acknowledge that while the intervention might be considered brief, it provides a preliminary assessment of feasibility and potential efficacy. Future studies could explore extended intervention durations to better assess the long-term impact and sustainability of the effects observed.
Comments 6: On the bottom of page 4 line 185 for you describe a “one-to-many format“ please provide additional details as to what that means
Response 6: Thank you for pointing this out.We agree that providing additional details will clarify the meaning of the "one-to-many format." In the revised manuscript, we have expanded the description of the "one-to-many format" used in our study.
Page 6 ,line 207: Specifically, this format refers to the structure where one facilitator (e.g., the mindfulness instructor) interacts with multiple participants simultaneously during the intervention sessions. The mindfulness sessions were conducted in a group setting, where the instructor guided the entire group of participants together.
Comments 7: Measurements. You need to use consistent verb tensing when discussing the various instruments utilized. For instance, you stated,
“2.5. Measurements
This study will utilize five scales, namely the Geriatric Depression Scale…” then later you stated, “Assessment of older adults' emotional states utilized three mood scales.” I am not sure what the verb tensing requirements are in the citation format you are using but minimally you should keep a consistent verb tensing for each of the various sections of the measurement tools utilized.
Response 7: Thank you for pointing out the inconsistency in verb tensing in the measurements section. We agree that consistent verb tense is important for clarity and coherence throughout the manuscript.We have reviewed the entire measurements section and revised the verb tensing to ensure consistency. Specifically, we have adjusted all references to the study design and use of scales to the past tense, as the data collection and assessments have already been completed.
Page 7 ,line 228: This study utilized five scales, namely the Geriatric Depression Scale 15-item version (GDS-15), the UCLA Loneliness Scale, the Brunel Mood Scale (BRUMS), the Satisfaction With Life Scale (SWLS), and the System Usability Scale (SUS).
Comments 8: Results. In addition to a very limited intervention, you also had very low numbers. If I am reading this correctly, your chat group had 15 participants, but only four completed the process. The mindfulness experiment had nine and eight completed the process. Those seem like very low numbers for making any kind of implications. It is interesting that in the chat group the four that completed the process are nearly 8 years older than the average for the 15 that started.
Response 8: Thank you for your detailed feedback. We acknowledge that the small sample sizes, particularly the low number of participants who completed the ChatGPT intervention, limit the generalizability of the findings.We acknowledge that the sample sizes of the ChatGPT group and mindfulness group were somewhat small and uneven due to participant compliance issues, which may limit the statistical power and generalisability of the findings. Specifically, some participants had difficulty maintaining regular attendance during the 8-week intervention, primarily due to scheduling conflicts and accessibility challenges. For example, some older adults interrupted the participation process due to health reasons, and some older adults arranged to meet with relatives and cancelled the participation. In the revised manuscript, we address this limitation in the discussion section. And suggests that future studies could benefit from a larger, more balanced sample.
Page 16, line 450: One of the limitations of the study is the relatively small and unbalanced sample sizes, with 15 participants in the ChatGPT group and 9 in the mindfulness group. This may limit the generalizability of the results. We observed several compliance-related challenges during the study. Some participants struggled to maintain consistent engagement with the 8-week intervention. For example, some participants stopped participating midway because they went to the hospital for treatment, and some participants refused to continue because they were not willing to express themselves. Future research could benefit from larger, more balanced samples to improve statistical power and the reliability of the findings.
Regarding the age difference, we agree that this is an interesting observation.We added a discussion section to draw attention to this age difference.
Page 16 ,line 447: It is interesting that in the chat group the four that completed the process are nearly 8 years older than the average for the 15 that started. Older participants may have had more time or motivation to complete the intervention, which could be a factor worth exploring in future studies.
Comments 9: “There were no significant between-group differences in any of the data, suggesting to some extent that there is no difference in effectiveness between ChatGPT chaperone chat and mindfulness practice.” P.7 lines 261-263. I can’t help but wonder if the fact that these were chaperoned chats had an impact here. That seems more of a social interaction than traditional AI chats. I realize that the population served in the study likely could not chat without assistance, but still, how much of this is the social experience of being with a chaperone.
Response 9: Thank you for raising this important point. We agree that the presence of a chaperone during the ChatGPT interactions could have influenced the results by adding a social component that might not be present in unassisted AI chats. The involvement of a chaperone, while necessary for some older adults in the study, may indeed have contributed to the overall emotional experience, potentially blending the effects of AI interaction with human social interaction.
Comments 10: I have a lot of concerns about your quantitative results since there was such a limited intervention for both the mindfulness and the chat AI experiences. My suggestion would be to increase the intervention significantly over a longer period of time, increase the number of participants, and look for ways to determine the actual causes of the changes rather than assuming they are simply from either the mindfulness or the chat AI experiences.
Response 10: We appreciate your valuable input. We agree that the limited duration of the intervention and the small sample size pose challenges to the robustness of our quantitative results.The adherence of participants has a large impact on the progress of the study, and we expect to improve this in future studies.
Comments 11: “Participants in both groups saw their life satisfaction scores rise after the intervention, and the upward trend was close to significant.” P.9 315–3 17. You cannot state that something is “close to significant” if it is not significant, then any changes could be based on things other than the intervention. Please remove that part of the sentence.
Response 11: Thank you for pointing out this issue. In response to your comment, we have revised the sentence to accurately reflect the statistical findings.We have removed that part of the sentence.
Page 10 ,line 282: In the Chat group, participants' 'happiness' emotions increased (3.25). Similarly, life satisfaction also increased (7.25). Life satisfaction in the Mindfulness group also increased (4.5).
Comments 12: “The disadvantages of the Chat group also exist, the experimental format is boring for the elderly. The dialog recognition requires more standard Mandarin, which is very difficult for the elderly.” P.9 330–332. This sounds like a pretty bold generalization. Perhaps you could state it as “some of the older adults involved in the chat group expressed…”
Response 12: Thank you for highlighting this issue. We agree that the original phrasing may overgeneralize the experiences of the participants.In response to your suggestion, we have revised the statement.
Page 14 ,line 363: Some of the older adults involved in the chat group expressed that the experimental format was less engaging and that the requirement for standard Mandarin in dialog recognition posed challenges.
Comments 13: “Elderly people like forms that imitate real beings, such as charming and cartoonish designs, like little animals.” page 9 lines 348–349. Here is another example of a bold generalization. In the line before that you say “elderly individuals typically communicate…“ At least that leaves the possibility that others may use other types of communication. Be careful about bold generalizations in your comments here. You cannot say all older adults do anything. You do not have the data to support that.
Response 13: Thank you for pointing out this concern regarding generalizations. In response to your feedback, we have revised the sentence to avoid broad generalizations.
Page 14 ,line 382: Some older adults may prefer forms that imitate real beings, such as charming and cartoonish designs, like little animals.
Comments 14: “Sight and hearing not only include appearance, but also mannerisms and tone of voice, and the elderly prefer to be in contact with real people” Page nine lines 351 to 352. Here is another example of generalization. Please go back through your results section and rephrase so you are not trying to indicate that all older adults have these experiences but only a subset of some of the people that you spoke with in this research.
Response 14: Thank you for pointing out this issue. In response to your feedback, we have revised the relevant section.
Page 14 ,line 385: Some participants indicated that sight and hearing include not only appearance but also mannerisms and tone of voice, and some expressed a preference for interacting with real people.
Comments 15: Your focus group results do not seem to be addressing the question at hand. I think there are many interesting pieces of information that you drew from the focus group, but as far as the question of the effectiveness of ChatGPT and mindfulness on minimizing depressive and loneliness symptoms, I really don’t feel like you got there with what you provided here.
Response 15: Thank you for your feedback regarding the focus group results.We understand that the focus group results may not have directly addressed the effectiveness of ChatGPT and mindfulness in reducing depressive and loneliness symptoms as clearly as intended. These qualitative insights help us understand the context and potential mechanisms behind the quantitative results. This broader perspective is valuable for interpreting the effectiveness of the interventions and identifying areas for future research.
Comments 16: The paragraph on page 10 from nine 395 to line 403 really provides an excellent overview of the limitations of your study. My suggestion would be to increase the intervention length both in the length of time for the individual interventions and the number of times the interventions occur. In other words, I think if you did this twice a week over a four month period of time and increased your participant size to include many more people you might have significant findings that would be much more valuable than what you have here.
Response 16: Thank you for your insightful suggestions. We appreciate your acknowledgment of the limitations discussed in our study. Your recommendation to extend the intervention duration and increase the frequency, as well as expanding the participant size, is valuable. Longer and more frequent interventions, coupled with a larger sample size, could indeed enhance the robustness and significance of the findings. These are important considerations for future research to improve the validity and generalizability of the results.
Round 2
Reviewer 1 Report
Comments and Suggestions for Authors
Thank you for the detailed reply to the revisions. All of my earlier recommendations have been met. The statistical analysis definitely needed a remake, which you did. Congratulations. However, there is still an issue with the citation 60. First author's name still appears as "SURNAME, A". Please correct this.
Author Response
Comments 1: There is still an issue with the citation 60. First author's name still appears as "SURNAME, A". Please correct this.
Response 1:Thank you for bringing this to our attention. We apologize for the oversight regarding citation 60. We have corrected the first author's name to ensure it appears properly in the revised manuscript.
Page 15, line 608: [60] HUANG G, HUI H, SHUNSHUN D, et al. The Effect of Social Participation on The Attitudes to Aging of Rural Elder-ly:Mediating Effects of Self-Efficacy and Loneliness [J]. Journal of Psychological Science, 2022, 45(4): 863-70.